# Structure, Antioxidant Activity and In Vitro Hypoglycemic Activity of a Polysaccharide Purified from *Tricholoma matsutake*

**DOI:** 10.3390/foods10092184

**Published:** 2021-09-14

**Authors:** Hui-Rong Yang, Lian-Hong Chen, Ying-Jie Zeng

**Affiliations:** College of Food Science & Technology, Southwest Minzu University, Chengdu 610041, China; zyjyhr@sina.com (H.-R.Y.); lianhong_chen@163.com (L.-H.C.)

**Keywords:** *Tricholoma matsutake*, polysaccharides, structure, antioxidant activity, hypoglycemic activity

## Abstract

The structure, antioxidant activity and hypoglycemic activity in vitro of a novel homogeneous polysaccharide from *Tricholoma matsutake* (Tmp) were investigated. Structural features suggested that Tmp was consisted of arabinose (Ara), mannose (Man), glucose (Glc) and galactose (Gal) with a molar ratio of 1.9:13.6:42.7:28.3, respectively, with a molecular weight of 72.14 kDa. The structural chain of Tmp was confirmed to contain →2,5)-α-l-Arabinofuranose (Ara*f*)-(1→, →3,5)-α-l-Ara*f*-(1→, β-d-Glucopyranose (Glc*p*)-(1→, α-d-Mannopyranose (Man*p*)-(1→, α-d-Galacopyranose (Gal*p*)-(1→, →4)-β-d-Gal*p*-(1→, →3)-β-d-Glc*p*-(1→, →3)-α-d-Man*p*-(1→, →6)-3-*O*-Methyl (Me)-α-d-Man*p*-(1→, →6)-α-d-Gal*p*-(1→, →3,6)-β-d-Glc*p*-(1→, →6)-α-d-Man*p*-(1→ residues. Furthermore, Tmp possessed strong antioxidant activity and showed the strong inhibitory effect on α-glucosidase and α-amylase activities. Then, a further evaluation found that there was a dramatic improvement in the glucose consumption, glycogen synthesis and the activities of pyruvate kinase and hexokinase when the insulin-resistant-human hepatoma cell line (IR-HepG2) was treated with Tmp. The above results indicated that Tmp had good hypoglycemic activity and also exhibited great potentials in in terms of dealing with type 2 diabetes mellitus.

## 1. Introduction

*Tricholoma matsutake* (*T. matsutake*) belongs to the *Subgenus Tricholoma* family, and is a traditional edible and medicinal fungus native chiefly to Asian countries, such as China, Japan, and Korea [1]. Moreover, *T. matsutake* has been applied to prevent and cure diseases such as diabetes and cardiovascular diseases for several thousand years [2]. In recent years, the bioactive natural product from those fungi identified as the homology of medicine and food have attracted increasing attention, mainly due to their non-toxic side effect [3,4,5,6]. There have been many published studies that reported that *T. matsutake* is rich in polysaccharides [1,2,7,8]. Moreover, polysaccharides from *T. matsutake* were confirmed to have anti-microorganism [1], anti-tumor [8,9], antioxidant [8] and immune activities [10]. Although there have been several polysaccharides isolated from *T. matsutake* with different structure and various bioactivities, little information could be available currently on their structure and in vitro hypoglycemic activity.

Nowadays diabetes mellitus (DM) has posed a great threat to the health of people all over the world. DM occurs due to metabolic disorders, especially when the insulin secreted by the islet cells is insufficient or the body could no longer utilize the insulin effectively, further resulting in the disorder of glucose metabolism [11]. Therefore, the targets that are selected for the research and development of the novel hypoglycemic drugs focus on those enzymes participating in the glucose metabolism. The most scientific approach to control the levels of postprandial blood glucose is to inhibit the activity of those enzymes involved in glucose metabolism such as α-amylase and α-glucosidase, further lowering down the formation of glucose [12]. The α-amylase and α-glucosidase inhibitors have the ability to decrease the carbohydrates, and then the absorption levels of the carbohydrate in the digestive tract decreased correspondingly [13]. There are many oral hypoglycemic medications now, such as acarbose, miglitol, thiazolidinediones, metformin etc., but the above medicines are synthetic, and may result in some side-effects and drug resistance [14]. Finding natural, effective and safe hyperglycemic compounds has become a matter of urgency. Recently, plenty of researchers are in broad agreement that natural polysaccharides exhibit antidiabetic activity [12,14,15,16,17].

Hence, the study aims to obtain a polysaccharide fraction from *T. matsutake*, and then elucidate the structural characteristics of the obtained polysaccharide fraction. Furthermore, the antioxidant activities such as hydroxyl and DPPH radical scavenging-capacities of the purified polysaccharide were evaluated. Finally, to assess the in vitro hypoglycemic activity, the inhibition effects and inhibition kinetics of the purified polysaccharide on the α-amylase and α-glucosidase were evaluated. Then, an insulin-resistant (IR)-human hepatoma cell line (HepG2) model was selected to illuminate the hypoglycemic activity in vitro of the obtained polysaccharide.

## 2. Materials and Methods

### 2.1. Materials and Chemicals

*T. matsutake* was picked from the “hometown of matsutake” in Yajiang County, Garze Tibetan Autonomous Prefecture, China. The fruiting bodies were cleaned with tap-water and then dried at 55 °C for 24 h, and then the samples were ground into powder and sieved through a 60-mesh sieve. All chemicals were of analytical grade.

### 2.2. Extraction and Purification of Crude Polysaccharides

The pigment in the *T. matsutake* powder were removed by ethanol (7:1 *w*/*v*). The solid–liquid ratio of *T. matsutake* powder and distilled water was 1:1.5. The extraction process was carried out with a 250 W ultrasonic power for 5 min. Then, the extract continued to be extracted for 2 h in boiling water bath. After centrifugation (3000× *g*, 15 min), the supernatant was collected and concentrated at 60 °C. The concentrated extract was further deproteinated by the Sevag reagent and AB-8 resin based on our previous literature [18]. After centrifugation, the mixture containing the solution and 95% ethanol with a volume ratio of 1:4 was kept at 4 °C for 12 h, and then removed the supernatant. The precipitate was lyophilized and the powder was the crude *T. matsutake* polysaccharides (TmpD). The TmpD was further purified by DEAE-52 and Sephadex G 200 based on our previous study [18].

### 2.3. Chemical Composition Analysis

The chemical compositions including the total carbohydrate content, protein content, and sulfate content were determined by the phenol-sulfuric acid method, the Lowry method and the method of barium chloride-gelatin nephelometry [19], respectively.

### 2.4. Structural Characteristics Analysis

The molecular weight (Mw) of purified *T. matsutake* polysaccharide (Tmp) were determined by HPGPC [19]. The monosaccharide composition was determined based on the previous method [16].

The methylation analysis and NMR analysis were conducted based on the published literature [20,21].

### 2.5. In Vitro Antioxidant Activity Evaluation

The DPPH radical-scavenging activity was analyzed based on the previous method [22]. The hydroxyl radical-scavenging activity was evaluated according to the previous literature [23].

### 2.6. Hypoglycemic Activity Analysis

The α-amylase inhibitory activity was investigated based on the reported method of Wang et al. [24]. The α-glucosidase inhibitory activity of Tmp was evaluated according to the previous literature [15].

To find out the inhibition type of Tmp on α-amylase and α-glucosidase, the enzyme inhibitory kinetics were performed based on the previous report [15].

### 2.7. In Vitro Hypoglycemic Assays

After the IR-HepG2 cells mode was constructed, the glucose consumption, the glycogen content, and the activities of hexokinase (HK) and pyruvate kinase (PK) were evaluated based on the previous literature [16].

### 2.8. Statistical Analysis

The data were presented as means ± standard deviation (SD). Analysis of variance (ANOVA) followed by one-way Dunnett’s test was performed using the statistical software package SPSS (SPSS Inc., Chicago, IL, USA), where *p* < 0.05 was considered statistically significant.

## 3. Results and Discussion

### 3.1. Physicochemical Property of Tmp

The yield of polysaccharides from *T. matsutake* occupied 3.23% of the dry powder weight. Although there were two polysaccharide fractions collected by DEAE-52 chromatography (Figure 1a), Tmp was the dominating polysaccharide fraction due to its high proportion, so Tmp was selected for further research. Furthermore, the single and symmetric curve of Tmp purified by Sephadex G 200 (Figure 1b) indicated that it was a homogeneous polysaccharide. UV spectra of Tmp showed no absorption peak in the wavelength of 260 nm and 280 nm (Figure 1c), which indicated that there were no impurity of nucleotides or protein. The contents of total carbohydrate, protein and sulfate in Tmp were exhibited in Table 1. The results suggested that Tmp accounted for 93.29 ± 1.79% of the total carbohydrate, which indicated that Tmp was the main ingredient exhibiting the biological activity. Meanwhile, there were a small number of sulfate content (6.87 ± 1.24%) and protein (6.92 ± 0.58%) in the Tmp fraction.

### 3.2. Monosaccharide Composition and Mw of Tmp

Based on the gas chromatography (GC) analysis of standard monosaccharides (Figure 2a), Tmp was consisted of the arabinose (Ara), mannose (Man), glucose (Glc) and galactose (Gal) with the molar ratio of 1.9:13.6:42.7:28.3, respectively (Figure 2b). The results suggested that the main molar ratio percentages in Tmp were Glc and Gal, and both Glc and Gal were the main compositional skeleton structure of Tmp. In addition, the Mw distribution of Tmp was exhibited in Figure 2c. The symmetrical and single peak suggested that Tmp was a homogeneous polysaccharide, and the Mw of Tmp was 72.14 kDa.

Interestingly, there were several studies focusing on the polysaccharides from *T. matsutake*, and the obtained polysaccharides with various monosaccharides exhibited many beneficial activities. For example, Cheng et al. had found that a homogeneous polysaccharide TMP-5II with the Mw of 15.76 kDa composed of D-Glc, D-Gal, D-Man and D-fucose [2]. You et al. had isolated three polysaccharides, TM-P1, TM-P2 and TM-P3, from *T. matsutake*. Structural analysis indicated that TM-P2 showed obvious difference from those of TMP1 and TM-P3. Furthermore, TM-P2 had the strongest in vitro antioxidant and anti-tumor activities [8]. Kim et al. found that polysaccharide β-glucan from pine-mushroom *T. matsutake* possessed immunostimulatory activities [10]. Based on the above previous studies, we could find that the monosaccharide composition and Mw of Tmp had great difference with the polysaccharides from *T. matsutake* in published literature. There was a broad consensus on the bioactivities of heteropolysaccharides that depended on the physicochemical property and structural characteristics [25,26]. Therefore, Tmp might show different biological activities.

### 3.3. Methylation Analysis of Tmp

Table 2 shows the results of glycosidic linkage analysis of Tmp. The characteristic iron fragments in mass spectrum spectra could reveal the glycosidic linkage types in Tmp [27]. According to the results summarized in Table 2, the (1→6) linked galacopyranose (Gal*p*) (36.51%) and (1→3) linked glucopyranose (Glc*p*) (25.23%) occupied the largest amount of residual in the backbone structure of Tmp. The presence of other glycosidic linkages suggested that there were branched residues in Tmp, that is, Tmp was branched. The molar ratios of 3-**Methyl (Me)_1_**-Arabinofuranose (Araf), 2-**Me_1_**-Ara*f*, 2,3,4,6-**Me_4_**-Glc*p*, 2,3,4,6-**Me_4_**-Mannopyranose (Manp), 2,3,4,6-**Me_4_**-Gal*p*, 2,3,6-**Me_3_**-Gal*p*, 2,4,6-**Me_3_**-Glc*p*, 2,4,6-**Me_3_**-Man*p*, 2,3,4-**Me_3_**-Man*p*, 2,3,4-**Me_3_**-Gal*p*, and 2,4-**Me_2_**-Glc*p* were 1.31:0.79:2.8:1.47:2.92:3.75:25.23:5.12:8.34: 36.51:12.08, respectively, which conformed with the overall monosaccharide composition described above.

### 3.4. NMR Spectroscopy Analysis

The chemical shifts in the ^1^H NMR and ^13^C NMR spectra (Figure 3a,b) were interpreted according to the results reported in the previous studies [18,28,29,30,31,32,33,34,35,36,37]. Meanwhile, the NMR spectra signals analysis were also combined with monosaccharide composition and glycosidic linkage analysis of Tmp (Table 3). The chemical shifts at δ 5.14, 5.04, 4.96, 4.90, 4.46, 5.04, 4.96, and 4.95 ppm were attributed to H-1 of →2,5)-α-l-Ara*f*-(1→, →3,5)-α-L-Ara*f*-(1→, α-d-Man*p*-(1→, α-d-Gal*p*-(1→, →4)-β-d-Gal*p*-(1→, →3)-α-d-Man*p*-(1→, →6)-3-*O*-Me-α-d-Man*p*-(1→, and →6)-α-d-Gal*p*-(1→ residues (Figure 3a). The corresponding chemical shifts from δ 3.2 to 4.0 ppm could be identified as sugar ring protons. In the ^13^C NMR spectrum, the main anomeric carbon signals were located at δ 99.53, 99.86, 101.09, 102.99, 102.90, 103.26, 104.54, 104.09, 107.25, and 107.95 ppm, corresponding to C-1 of →6)-α-d-Gal*p*-(1→, α-d-Gal*p*-(1→, →6)-α-d-Man*p*-(1→, α-d-Man*p*-(1→, →3)-α-d-Man*p*-(1→, β-d-Glc*p*-(1→, →3)-β-d-Glc*p*-(1→, →3,6)-β-d-Glc*p*-(1→, →3,5)-α-l-Ara*f*-(1→ and →2,5)-α-l-Ara*f*-(1→ residues (Figure 3b). The carbon signals located at δ 60~86 ppm could be assigned to the downfield shift due to the substitutions in C-2, C-3 or C-4. In addition, the chemical shifts at δ 61~65 ppm could be ascribed to the C-6. The shift in peaks at δ 67.74, 67.69, 70.27, 70.93, and 70.27 ppm to low field suggested that there were the substitutions at C-6. The above results further indicated that there were glucosyl, mannosyl and galactosyl residues in Tmp.

According to the above analysis, the whole assignments of the linkage types in Tmp were shown in Table 3. Above all, Tmp was determined to be mainly consisted of →2,5)-α-l-Ara*f*-(1→, →3,5)-α-l-Ara*f*-(1→, β-d-Glc*p*-(1→, α-d-Man*p*-(1→, α-d-Gal*p*-(1→, →4)-β-d-Gal*p*-(1→, →3)-β-d-Glc*p*-(1→, →3)- α-d-Man*p*-(1→, →6)-3-*O*-Me-α-d-Man*p*-(1→, →6)-α-d-Gal*p*-(1→, →3,6)-β-d-Glc*p*-(1→, →6)-α-d-Man*p*-(1→ residues.

### 3.5. Antioxidant Activity

Hydroxyl radicals are reactive oxygen species (ROS) that are crucial to maintain the steady-state balance of normal physiological metabolism in the body. Natural polysaccharides have been proven to clean or inhibit the production of ROS [38,39,40]. Although the ascorbic acid exhibited a higher scavenging rate than Tmp to the hydroxyl radicals (Figure 4a), Tmp still showed high removal efficiency on hydroxyl and presented a dose-dependent effect. Additionally, 0.8 mg/mL of Tmp could reach up to 68.74% of the hydroxyl removal efficiency, while the positive group, ascorbic acid, was 96.52% at the same concentration. Published literature had indicated that those polysaccharides with sulfate groups, low or moderate Mw might possess strong antioxidant activity [19,41]. Interestingly, the strong hydroxyl scavenging capacity of Tmp could be attributed to the sulfate groups and low MW.

As for Tmp, the higher of the concentration was, the greater of the radical-scavenging rates of Tmp increased compared with the positive control ascorbic acid (Figure 4b). That is, the radical-scavenging rates of Tmp appeared in concentration-dependence. When the content of Tmp was 0.4 mg/mL, the radical-scavenging rate reached up to 94.82%, which suggested that Tmp possessed better antioxidant activity and had stronger ability to donate electrons or hydrogen.

### 3.6. Inhibitory Effect of Tmp on the α-Amylase and α-Glucosidase

The α-amylase and α-glycosidase, like other digestive enzymes, were able to break down the dietary carbohydrates in small intestines to generate mono- and/or disaccharides [42]. Hence, how to inhibit the activity of α-amylase and α-glycosidase has been a promising approach to prevent hyperglycemia. In this study, various concentrations of Tmp (0.75~12 mg/mL) possessed an inhibitory effect on the α-amylase activity and presented a concentration-dependent manner (Figure 5a). Moreover, the IC_50_ value of Tmp was 3.75 mg/mL, and the IC_50_ value of Tmp was obviously more than that of the positive group acarbose (12.15 μg/mL). However, Tmp remained to be considered as a potential candidate of hypoglycemic drug.

Lineweaver–Burk plot was used as the mode to evaluate the inhibition effect of Tmp on α-amylase activity. According to Figure 5b, the *K_m_* and *V_max_* values decreased as Tmp concentration increased. Moreover, an uncompetitive manner of Tmp on α-amylase was observed due to the nearly unchanged slopes of three inhibition kinetics curves (*K_m_*/*V_max_*), which indicated that there was no competition between Tmp and its substrate. The subjective reason in appearance of the above phenomenon is put down to the physical interference effect of polysaccharides [43]. Based on the published literature, there have rarely been reports on the natural uncompetitive inhibitors of α-amylase, but it has almost no effect on their advantages in developing medicaments [44].

The inhibition effect of Tmp on the α-glucosidase was presented in Figure 5c. The results indicated that the inhibition action enhanced in a concentration-depend manner. The value of IC_50_ was recorded as 46-fold lower than that of the positive group acarbose (0.1 mg/mL vs. 4.63 mg/mL). The above significant difference portended that there were more inhibitory groups or components in Tmp targeting the active sites of α-glucosidase. Meanwhile, the results suggested that compared to acarbose, Tmp was identified as a candidate with great potential to inhibit the α-glucosidase activity and further trials in vivo and in vitro on hyperglycemic mode remained to be done.

It showed that the inhibitory model of Tmp on α-glucosidase was the double-reciprocal Lineweaver–Burk plots (Figure 5d). Three plot lines with various slopes would intersect. The V_max_ and K_m_ lowered down while the concentrations of Tmp enhanced, which demonstrated that Tmp played a mixed inhibition effect on α-glucosidase. Above all, the results portended that Tmp exhibited great potential to an inhibitor of α-glucosidase.

### 3.7. The Consumption of Glucose Analysis

IR did not occur until target cells/tissues reduced uptake and utilization of glucose, further causing abnormal blood glucose homeostasis [45]. When the IR-HepG2 cell was treated with Tmp, there was an obvious decrease, from 5.79 mM to 1.87 mM, in the content of glucose consumption compare to the positive group (Figure 6a). The above results suggested that the established model was successful. Furthermore, based on Figure 5a, the amount of glucose consumption in the control group increased by 5.82 mM. The glucose consumption in model group enhanced and presented in a dose-depend manner, and the IR-HepG2 cells treated with 2.0 mg/mL of Tmp corresponded to the most glucose consumption (5.18 mM). Interestingly, there were many published studies obtaining similar results. Ren at al. [22] had reported that polysaccharides STP-1 from *Sargassum thuand* contributed to improving the glucose uptake in IR-HepG2 cells. Liu et al. [46] found that *Monostroma angicava* Kjellm polysaccharide presented a stimulative effect on the glucose consumption of IR-HepG2 cells.

### 3.8. Intracellular Enzyme Activity and Glycogen Content Analysis

The occurrence of IR would cause the down-regulation of related enzyme expression of the hepatic glycolytic process, such as hexokinase (HK) and pyruvate kinase (PK), finally resulting in the decrease of glucose utilization [47]. Therefore, the HK and PK in IR-HepG2 cells treated with Tmp were selected to investigate the modulating effect of Tmp on the glucose metabolism. As exhibited in Figure 6b,c, a significant increase of the HK and PK activities was observed in the sample-treated group. While the IR-HepG2 cells were treated with Tmp, the HK and PK activities almost reached up to the level of the positive groups (8.65 U/g protein vs. 8.27 U/g protein for HK, 44.91 mg/g protein vs. 46.35 mg/g protein for PK), which indicated that there were positive effects of Tmp on the increase of the glycolysis and glucose metabolism. Coincidentally, this similar phenomenon had been reported in previous study that polysaccharide LBP-s-1 from *Lycium barbarum* L. presented strong effect on the HK and PK activities in the hepatic of diabetic animals, suggesting that LBP-s-1 possessed significant hypoglycemic effect [48]. Therefore, Tmp might be identified as a hypoglycemic functional component with the capacities of increasing glucose utilization, HK and PK.

Additionally, the analysis of intracellular glycogen content indicated that the glycogen level in the model group exhibited markedly decrease compared with the normal group (Figure 6d). However, when the IR-HepG2 mode group was treated with Tmp, the level of glycogen gradually restored in a concentration-dependent manner, indicating that Tmp might improve the heptatic insulin sensitivity and then facilitate the synthesis of glycogen [49].

## 4. Conclusions

In this study, a novel and homogeneous polysaccharide Tmp from *T. matsutake* was obtained and the structural features were characterized. Tmp was mainly consisted of Ara, Man, Glc, and Gal with a molar ratio of 1.9:13.6:42.7:28.3, respectively. The main structure of Tmp was composed of →2,5)-α-l-Ara*f*-(1→, →3,5)-α-l-Ara*f*-(1→, β-d-Glc*p*-(1→, α-d-Man*p*-(1→, α-d-Gal*p*-(1→, →4)-β-d-Gal*p*-(1→, →3)-β-d-Glc*p*-(1→, →3)- α-d-Man*p*-(1→, →6)-3-*O*-Me-α-d-Man*p*-(1→, →6)-α-d-Gal*p*-(1→, →3,6)-β-d-Glc*p*-(1→, →6)-α-d-Man*p*-(1→ residues. Moreover, Tmp showed good antioxidant activity, α-amylase and α-glucosidase inhibition activity. The inhibition kinetics analysis indicated that Tmp exhibited an uncompetitive manner on α-amylase, but played a mixed inhibition effect on α-glucosidase. Hypoglycemic tests in IR-HepG2 cell mode revealed that Tmp possessed favorable hypoglycemic activity. Further study on hypoglycemic activity needs to be carried out in vivo studies for optimization.

## Figures and Tables

**Figure 1 foods-10-02184-f001:**
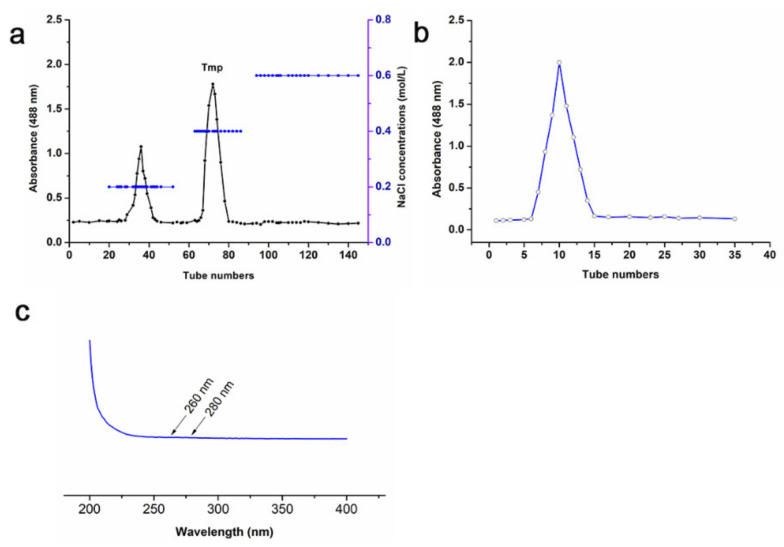
Chromatography of Tmp purified by DEAE-52 chromatography (**a**) and Sephadex G-200 (**b**); UV spectrum of Tmp (**c**).

**Figure 2 foods-10-02184-f002:**
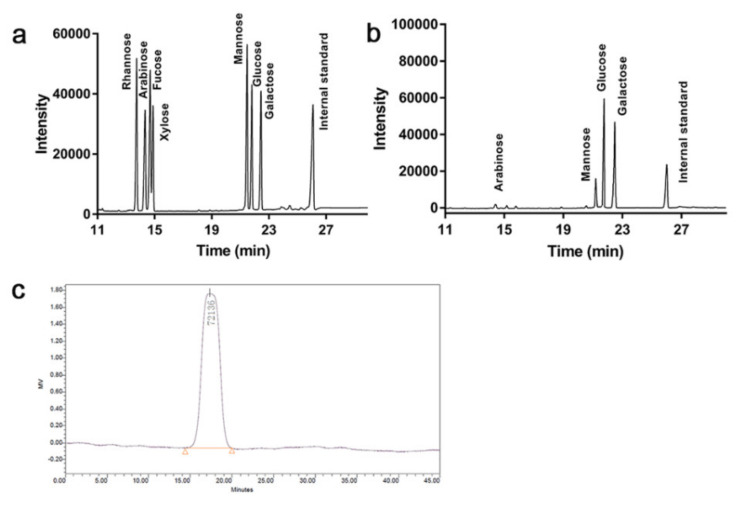
GC of standard monosaccharides (**a**) and Tmp (**b**); HPGPC of Tmp (**c**).

**Figure 3 foods-10-02184-f003:**
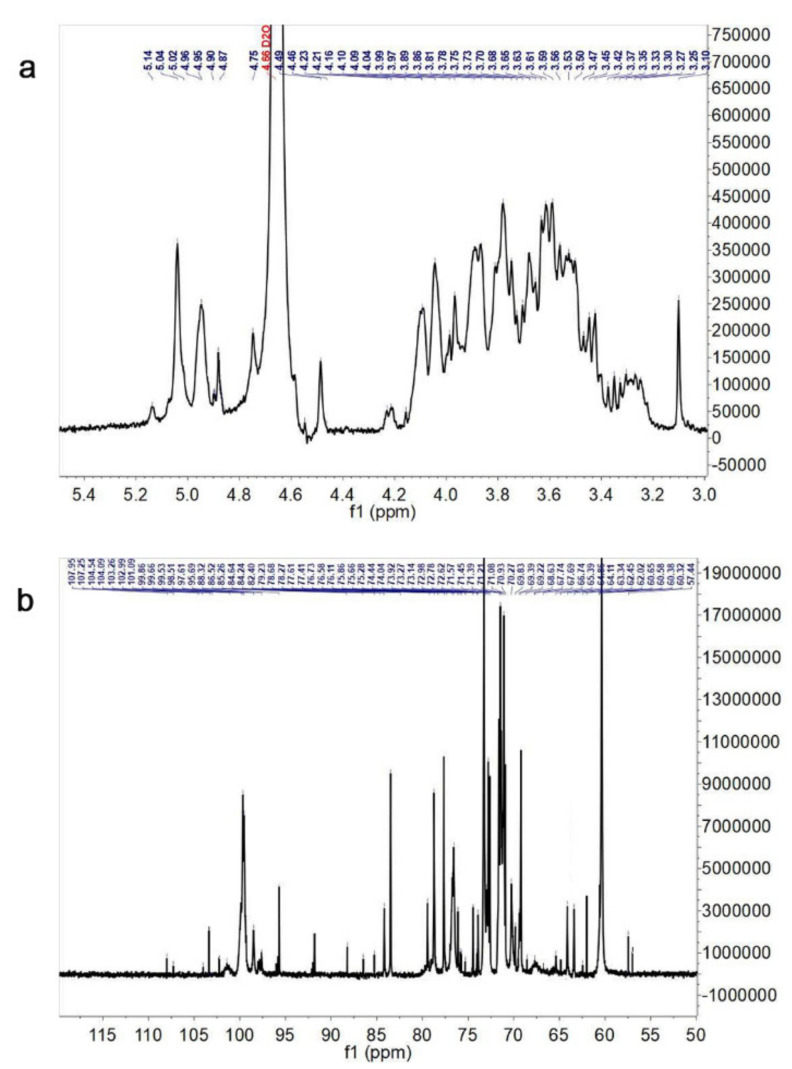
^1^H NMR spectrum (**a**) and ^13^C NMR spectrum (**b**) of Tmp.

**Figure 4 foods-10-02184-f004:**
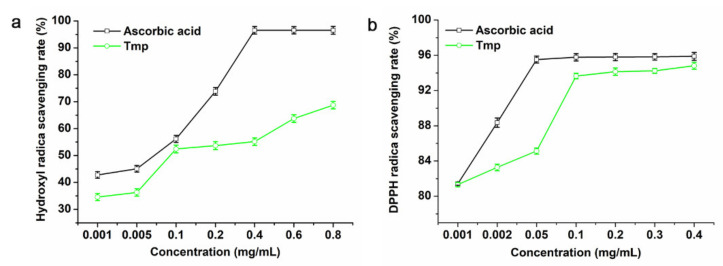
Scavenging activities of Tmp on hydroxyl radicals (**a**) and DPPH radicals (**b**).

**Figure 5 foods-10-02184-f005:**
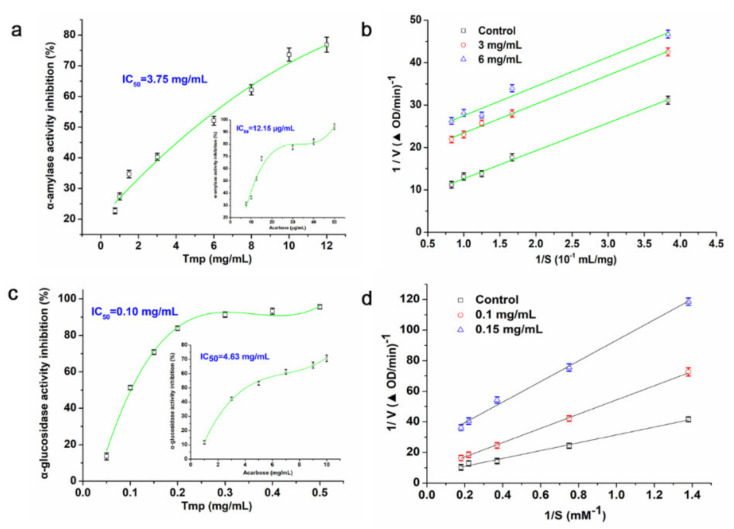
Inhibitory effect of Tmp on α-amylase activity (**a**). Inhibition kinetics of Tmp on α-amylase activity (**b**). Inhibitory effect of Tmp on α-glucosidase activity (**c**). Inhibition kinetics of Tmp on α-glucosidase activity (**d**). Acarbose was used as the positive control.

**Figure 6 foods-10-02184-f006:**
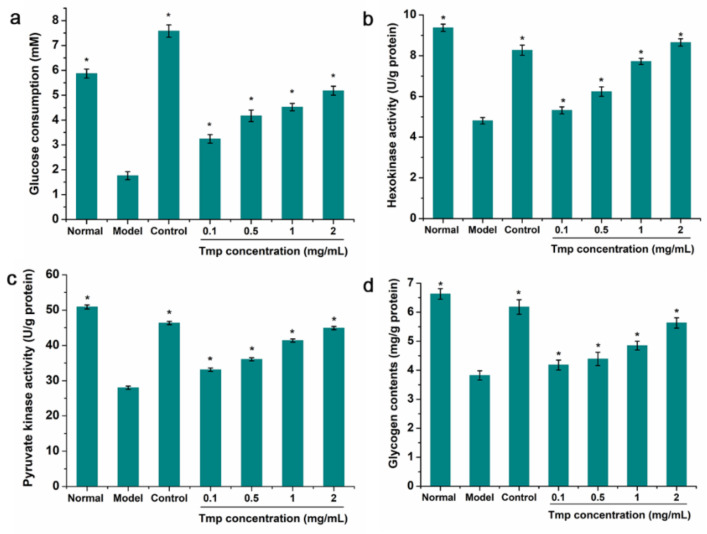
Effects of Tmp with various concentrations on glucose consumption (**a**), HK activity (**b**), PK activity (**c**) and glycogen content (**d**) in IR-HepG2 cells. Metformin (3.5 mg/mL) was used as the control group. * *p* < 0.05 vs. the model group. Values are presented as mean ± SD of triplicate experiments.

**Table 1 foods-10-02184-t001:** Chemical composition of Tmp.

	Carbohydrate Content (%)	Protein Content (%)	Sulfate Content (%)
Tmp	93.29 ± 1.79	6.92 ± 0.58	6.87 ± 1.24

**Table 2 foods-10-02184-t002:** Methylation analysis and mode of linkage of Tmp.

Methylated Sugar Resides	Mass Fragments (m/z)	Linkage Types	Molar Ratio (%)
3-**Me****_1_**-Ara*f*	43, 71, 87, 129, 189	→2,5)-Araf-(1→	1.31
2-**Me****_1_**-Ara*f*	43, 71, 85, 99, 115, 117, 127	→3,5)-Araf-(1→	0.79
2,3,4,6-**Me****_4_**-Glc*p*	43, 71, 87, 101, 117, 129, 145, 161, 205	Glc-(1→	2.8
2,3,4,6-**Me****_4_**-Man*p*	43, 71, 87, 101, 117, 129, 145, 161, 205	Man-(1→	1.47
2,3,4,6-**Me****_4_**-Gal*p*	43, 71, 87, 101, 117, 129, 145, 161, 205	Gal-(1→	2.92
2,3,6-**Me_3_**-Gal*p*	43, 87, 99, 101, 113, 117, 129, 131, 161, 173, 233	→4)-Galp-(1→	3.75
2,4,6-**Me****_3_**-Glc*p*	43, 71, 85, 87, 99, 101, 117, 129, 161	→3)-Glc-(1→	25.23
2,4,6-**Me****_3_**-Man*p*	43, 71, 85, 87, 99, 101, 117, 129, 161	→3)-Man-(1→	5.12
2,3,4-**Me****_3_**-Man*p*	43, 71, 87, 99, 101, 117, 129, 161, 173, 189, 233	→6)-3-O-Me-Man-(1→	8.34
2,3,4-**Me****_3_**-Gal*p*	43, 71, 87, 99, 101, 117, 129, 161, 173, 189, 233	→6)-Gal-(1→	36.51
2,4-**Me****_2_**-Glc*p*	43, 71, 87, 99, 101, 117, 129, 161, 173, 189	→3,6)-Glcp-(1→	12.08

**Table 3 foods-10-02184-t003:** ^1^H and ^13^C NMR chemical shifts of Tmp (δ, ppm).

Resides	H1/C1	H2/C2	H3/C3	H4/C4	H5/C5	H6/C6	O-Me
→2,5)-α-L-Ara*f*-(1→	5.14/107.95	4.09/88.32	4.16/76.93	3.97/84.24	3.81/67.74	3.63/—	
→3,5)-α-L-Ara*f*-(1→	5.04/107.25	4.04/82.40	4.04/84.69	3.75/79.23	3.59/67.69	3.81/—	
β-d-Glc*p*-(1→	4.75/103.26	3.33/70.27	3.63/75.3	3.81/70.27	ND ^1^	3.86/62.02	
α-d-Man*p*-(1→	4.96/102.99	4.04/71.57	3.81/72.23	3.37/75.28	3.61/68.63	3.81/63.34	
α-d-Gal*p*-(1→	4.90/99.86	3.78/71.45	4.16/70.17	3.61/74.44	ND	3.61/64.11	
→4)-β-d-Gal*p*-(1→	4.46/103.26	3.42/74.04	3.61/73.88	4.04/78.27	3.53/74.44	3.73/64.86	
→3)-β-d-Glc*p*-(1→	4.75/104.54	3.47/74.44	3.70/85.78	3.45/69.83	3.42/76.73	3.81/62.02	
→3)- α-d-Man*p*-(1→	5.04/102.99	4.16/70.93	3.99/78.23	3.78/68.63	3.81/75.86	3.73/62.45	
→6)-3-*O*-Me-α-d-Man*p*-(1→	4.96/103.26	3.81/71.39	3.81/79.3	4.21/66.74	3.53/74.44	3.78/70.27	3.37/57.44
→6)-α-d-Gal*p*-(1→	4.95/99.53	3.56/72.62	3.78/71.89	ND	ND	3.75/70.93	
→3,6)-β-d-Glc*p*-(1→	4.49/104.09	3.30/74.44	3.68/85.97	3.30/70.93	3.42/77.41	3.81/70.27	
→6)-α-d-Man*p*-(1→	4.49/101.09	3.35/72.62	3.63/72.62	3.70/68.63	3.97/71.57	3.78/70.93	

^1^ ND, signal was not detected.

## Data Availability

Data are contained within the article.

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
