# Peer review of "Structure, Antioxidant Activity and In Vitro Hypoglycemic Activity of a Polysaccharide Purified from Tricholoma matsutake"

_foods, 2021, doi:10.3390/foods10092184_

Round 1

Reviewer 1 Report

The authors presents a work on the structure and several properties of a polysaccharide obtained from the Tricholoma matsutake fungus. The paper  is interesting and the information given by the authors are supported by an extensive bibliography. Even with these considerations, the paper in its current form should be revised due to the presence of many inaccuracies and errors such as the correction of some typos and figures, and the addition of some information that can help the reader.
Below I propose a list of different comments/corrections to be made in the article, referring to the proof copy.

  1. Abstract Section (remark): it is suggested not to use abbreviations in the abstract or if they are required to define them better if repeated several times [e.g., Tmp (Tricholoma matsutake polysacch.), Me (Methyl), Araf (Arabinose binding protein?), Glcp (Glucopyranose), etc.].
  2. Introduction Section (remark): precisely due to the fact that abbreviations should not be used in the abstract, they must be properly defined in the introduction of the article. 
  3. Introduction Section, page 2, line 51 (typo): eliminate the comma, "... exhibit antidiabetic activity [...".
  4. Materials and Methods Section (remark): even if there are several references to the techniques used, I suggest to indicate all the methodologies or, at least, the instruments used and in short the experimental methodologies to help the reader in understanding the experiment and in its possible reproduction.
  5. Materials and Methods Section, page 2, line 90 (typo): I suggest to modify the hydroxyl radical notation in ·OH.
  6. Results and Discussion Section, page 3, lines 104-113 (remarks): there are no references in the text to the figures 1a, 1b and 1c; furthermore, I advise you to deepen the description of the aforementioned figures.
  7. Results and Discussion Section, page 4, lines 119-138 (remarks): in this paragraph there is a lot of confusion about the position and description of the figures, in addition to the molar ratio values; for example:
    1. Fig. 2a: As stated by the authors in the caption of figure 2, fig. 2a represents the gas chromatography analysis of standard monosaccharides; so, why it refers to Tmp analysis in the text? (line 119) -> I suggest to insert a properly description of this figure in the text connecting it with figure 2b that describe Tmp, otherwise it is preferable to remove it.
    2. Molar Ratio: there is an important error/typo regarding the position of glucose and mannose in the list of molar ratio. Please, check it.
  8. Results and Discussion Section, page 4-5, lines 143-152 (remark): MS -> mass spectrum; all the abbreviations to be introduced as well as the subscripts of the methylation analysis should be deepened to help the reader. 
  9. Results and Discussion Section, page 5-6, lines 155-181 (remark): as before, the figures are wrongly described in the caption and then wrongly inserted or even not inserted in the text. Fig. 3a -> 13C and Fig. 3b -> 1H. Please, give a proper check to this issue. 
  10. Results and Discussion Section, page 6, lines 183 (typos/remarks): I suggest to give a detailed description of Reactive Oxygen Species (ROS), to check (again) the figure cited in the text (line 185) and to modify the hydroxyl radical notation in ·OH (line 190 and 192). 
  11. Results and Discussion Section, page 9, lines 249-275 (remarks):
    the use of the p-value (line 252) is debated [Nature, 567, 305-307 (2019)], so I suggest to insert a brief description in the Materials and Methods Section of why that specific value was considered and about the lower and upper limit of the intervals considered; furthermore, the figures were again incorrectly indicated or not described in the text (see line 259).
  12. References Section: the authors are asked to carefully check this section as there are several indications of volumes, pages, etc. (see lines: 321, 324, 326, 352, 382, 402).

Author Response

Dear Editor/Reviewers,

We are very grateful to you for your consideration of our manuscript (Manuscript ID: foods-1378908) for publication in Foods and helpful suggestions for revision. We have revised the manuscript carefully according to the reviewers’ comments and the editor’s suggestions. The issues raised by the reviewers have been addressed as follows. Page, line and figure numbers refer to the revised manuscript.

  1. Abstract Section (remark): it is suggested not to use abbreviations in the abstract or if they are required to define them better if repeated several times [e.g., Tmp (Tricholoma matsutake polysacch.), Me (Methyl), Araf (Arabinose binding protein?), Glcp (Glucopyranose), etc.].

Response: Thanks for the referee's helpful suggestion. As suggested by the referee, we have used the full names in the abstract section. The description is as follows:

“Abstract: The structure, antioxidant activity and hypoglycemic activity in vitro of a novel ho-mogeneous polysaccharide from Tricholoma matsutake (Tmp) were investigated. Structural features suggested that Tmp was consisted of arabinose (Ara), mannose (Man), glucose (Glc) and galactose (Gal) with a molar ratio of 1.9:13.6:42.7:28.3, respectively. with a molecular weight of 72.14 kDa The structural chain of Tmp was confirmed to contain 2,5)-α-L-Arabinofuranose (Araf)-(1, 3,5)-α-L-Araf-(1, β-D-Glucopyranose (Glcp)-(1, α-D-Mannopyranose (Manp)-(1, α-D-Galacopyranose (Galp)-(1, 4)-β-D-Galp-(1, 3)-β-D-Glcp-(1, 3)-α-D-Manp-(1, 6)-3-O-Methyl (Me)-α-D-Manp-(1, 6)-α-D-Galp-(1, 3,6)-β-D-Glcp-(1, 6)-α-D-Manp-(1 residues. Furthermore, Tmp possessed strong antioxidant activity and showed the strong inhibitory effect on α-glucosidase and α-amylase activities. Then a further evaluation found that there was a dramatically improvement in the glucose consumption, glycogen synthesis and the activities of pyruvate kinase and hexokinase when the insulin-resistant-human hepatoma cell line (IR-HepG2) were treated with Tmp. The above results indicated that Tmp had good hypoglycemic activity and also exhibited great potentials in in terms of dealing with Type 2 diabetes mellitus.”

  1. Introduction Section (remark): precisely due to the fact that abbreviations should not be used in the abstract, they must be properly defined in the introduction of the article.

Response: Thanks for the referee's helpful suggestion. As suggested by the referee, we have properly defined the abbreviations in the introduction of the article.

  1. Introduction Section, page 2, line 51 (typo): eliminate the comma, "... exhibit antidiabetic activity [...".

Response: Thanks for the referee's helpful suggestion. As suggested by the referee, we have eliminated the comma in the end of the sentence "... exhibit antidiabetic activity [...".

  1. Materials and Methods Section (remark): even if there are several references to the techniques used, I suggest to indicate all the methodologies or, at least, the instruments used and in short the experimental methodologies to help the reader in understanding the experiment and in its possible reproduction.

Response: Thanks for the referee's helpful suggestion. As suggested by the referee, we have added the detail experimental methodologies in the Supplementary Materials.

  1. Materials and Methods Section, page 2, line 90 (typo): I suggest to modify the hydroxyl radical notation in ·OH.

Response: Thanks for the referee's helpful suggestion. As suggested by the referee, we have modified the expression of ·OH in page 2, line 90 (typo) as “hydroxyl radical-scavenging activity”.

  1. Results and Discussion Section, page 3, lines 104-113 (remarks): there are no references in the text to the figures 1a, 1b and 1c; furthermore, I advise you to deepen the description of the aforementioned figures.

Response: Thanks for the referee's helpful suggestion. As suggested by the referee, we have added the figures 1a, 1b and 1c in the text (page 3, lines 104-113), and supplemented the description of figure 1c “UV spectra of Tmp showed no absorption peak in the wavelength of 260 nm and 280 nm (Figure 1c), which indicated that there were no impurity of nucleotides or protein”. Furthermore, we have deepened the description of the aforementioned figures.

  1. Results and Discussion Section, page 4, lines 119-138 (remarks): in this paragraph there is a lot of confusion about the position and description of the figures, in addition to the molar ratio values; for example:
  2. Fig. 2a: As stated by the authors in the caption of figure 2, fig. 2a represents the gas chromatography analysis of standard monosaccharides; so, why it refers to Tmp analysis in the text? (line 119) -> I suggest to insert a properly description of this figure in the text connecting it with figure 2b that describe Tmp, otherwise it is preferable to remove it.

Response: Thanks for the referee's helpful suggestion. As suggested by the referee, we have modified the description of the figure 2a and 2b, because the monosaccharide composition of Tmp (Figure 2a) was obtained based on the gas chromatography analysis of standard monosaccharides (Figure 2a).

The description is as follows:

“Based on the gas chromatography (GC) analysis of standard monosaccharides (Figure 2a), Tmp was consisted of the arabinose, mannose, glucose and galactose with the molar ratio of 1.9:13.6:42.7:28.3, respectively (Figure 2b).” 

  1. Molar Ratio: there is an important error/typo regarding the position of glucose and mannose in the list of molar ratio. Please, check it.

Response: Thanks for the referee's helpful suggestion. As suggested by the referee, we have modified the order of molar ratio as follows:

“Tmp was consisted of the arabinose, mannose, glucose and galactose with the molar ratio of 1.9:13.6:42.7:28.3, respectively”.

  1. Results and Discussion Section, page 4-5, lines 143-152 (remark): MS -> mass spectrum; all the abbreviations to be introduced as well as the subscripts of the methylation analysis should be deepened to help the reader.

Response: Thanks for the referee's helpful suggestion. As suggested by the referee, we have modified MS as mass spectrum in page 4-5, lines 143-152.

The description is as follows:

The characteristic iron fragments in mass spectrum spectra······

Furthermore, as suggested by the referee, we have introduced all the abbreviations and the subscripts of the methylation analysis have been deepened to help the reader.

The description is as follows:

Table 2 was the results of glycosidic linkage analysis of Tmp. The characteristic iron fragments in mass spectrum spectra could revealed the glycosidic linkage types in Tmp [28]. According to the results summarized in Table 2, the (16) linked galacopyranose (Galp) (36.51%) and (13) linked glucopyranose (Glcp) (25.23%) occupied the largest amount of residual in the backbone structure of Tmp. The presence of other glycosidic linkages suggested that there were the branched residues in Tmp, that is, Tmp was branched. The molar ratios of 3-Methyl (Me)1-Arabinofuranose (Araf), 2-Me1-Araf, 2,3,4,6-Me4-Glcp, 2,3,4,6-Me4-Mannopyranose (Manp), 2,3,4,6-Me4-Galp, 2,3,6-Me3-Galp, 2,4,6-Me3-Glcp, 2,4,6-Me3-Manp, 2,3,4-Me3-Manp, 2,3,4-Me3-Galp, and 2,4-Me2-Glcp were 1.31:0.79:2.8:1.47:2.92:3.75:25.23: 5.12:8.34: 36.51:12.08, respectively, which conformed with the overall monosaccharide composition described above.

  1. Results and Discussion Section, page 5-6, lines 155-181 (remark): as before, the figures are wrongly described in the caption and then wrongly inserted or even not inserted in the text. Fig. 3a -> 13C and Fig. 3b -> 1H. Please, give a proper check to this issue.

Response: Thanks for the referee's helpful suggestion. As suggested by the referee, we have adjusted the order of 1H and 13C NMR spectra. Meanwhile, we have checked all the similar issues and made changes in the revised manuscript.

  1. Results and Discussion Section, page 6, lines 183 (typos/remarks): I suggest to give a detailed description of Reactive Oxygen Species (ROS), to check (again) the figure cited in the text (line 185) and to modify the hydroxyl radical notation in ·OH (line 190 and 192).

Response: Thanks for the referee's helpful suggestion. As suggested by the referee, we have given a detailed description of ROS, checked the figure cited in the text and modified the hydroxyl radical notation.

The description is as follows:

“Hydroxyl radicals are crucial reactive oxygen species (ROS) to maintain the steady-state balance of normal physiological metabolism in body.” 

  1. Results and Discussion Section, page 9, lines 249-275 (remarks): the use of the p-value (line 252) is debated [Nature, 567, 305-307 (2019)], so I suggest to insert a brief description in the Materials and Methods Section of why that specific value was considered and about the lower and upper limit of the intervals considered; furthermore, the figures were again incorrectly indicated or not described in the text (see line 259).

Response: Thanks for the referee's helpful suggestion. As suggested by the referee, we have added the section of Statistical analysis as follows:

“2.8 Statistical analysis

The data were presented as means ± standard deviation (SD). Analysis of variance (ANOVA) followed by one-way Dunnett’s test was performed using the statistical soft-ware package SPSS (SPSS Inc., Chicago, IL, USA), where p < 0.05 was considered statisti-cally significant.

Furthermore, due to our careless, we have corrected the chart numbers in the revised manuscript, e.g. in page 9, lines 249-275, Figure 6a, 6b, 6c and 6d have replaced Figure 5a, 5b, 5c and 5d.

  1. References Section: the authors are asked to carefully check this section as there are several indications of volumes, pages, etc. (see lines: 321, 324, 326, 352, 382, 402).

Response: Thanks for the referee's helpful suggestion. As suggested by the referee, we have checked the references and made changes according to the Instructions for Authors. 

Best regards,

Ying-Jie Zeng, Associate Professor, Corresponding author

College of Food Science & Technology,

Southwest Minzu University,

Wuhou District, Chengdu, China

E-mail: yjzeng@swun.edu.cn; phhzyj@163.com

Cell: +86-15602408766

Reviewer 2 Report

The authors are kindly recommended to check the attached comments of the reviewer

Author Response

Dear Editor/Reviewers,

We are very grateful to you for your consideration of our manuscript (Manuscript ID: foods-1378908) for publication in Foods and helpful suggestions for revision. We have revised the manuscript carefully according to the reviewers’ comments and the editor’s suggestions. The issues raised by the reviewers have been addressed as follows. Page, line and figure numbers refer to the revised manuscript.

(I) Introduction

  • Line 34: proposed reformulation: instead of “could be” to consider “ currently…”?
  • Lines 36: proposed reformulation: instead of “now” to consider “nowadays”?

Response: Thanks for the referee's helpful suggestion. As suggested by the referee, we have added currently” in the sentence as follows: “little information could be available currently on their ……” in Line 34 and “now” has been replaced with  “nowadays” in Line 36.

(II) Material & Methods

Please include a section with Statistical analysis. You give some rough info in tables/figures but not clear picture of data analysis. For instance: (i) authors provide SD values but do not refer on basis of which Nr of replicates? (n=?) (ii) figure -6: how did authors decide for statistically significant differences. Any ANOVA testing to conclude statistical differences of means at p<0.05?  Please indicate here as well the order of statistically different activities in letters (e.g. a>b>c…?)

Response: Thanks for the referee's helpful suggestion. As suggested by the referee, we have added the section of Statistical analysis as follows:

“2.8 Statistical analysis

The data were presented as means ± standard deviation (SD). Analysis of variance (ANOVA) followed by one-way Dunnett’s test was performed using the statistical soft-ware package SPSS (SPSS Inc., Chicago, IL, USA), where p < 0.05 was considered statisti-cally significant.

(III) Results & discussions

  • Lines 135-136:” we could find that the monosaccharide composition and Mw 134 of Tmp had great difference with the polysaccharides from T. matsutake in published literature….” Any explanation for this observation (e.g. extraction or analytical methods?)
  • Section 3.5. “antioxidant activity”: The authors refer to concentration dependent effect for antioxidant activity based on OH-scavenging activity method…however, this is not the case for DPPH method where antioxidant activity start reaching a “plateau” after 0.1 mg/ml…please state that explicitly to inform that this observation is also method-dependent. Any similar findings in literature to back up such observations?

Response: Thanks for the referee's helpful suggestion. As for       Lines135-136:” we could find that the monosaccharide composition and Mw of Tmp had great difference with the polysaccharides from T. matsutake in published literature….”, we draw the conclusion based on the comparison of monosaccharide composition and molecular weight of the published literatures focusing on others polysaccharides from T. matsutake and Tmp obtained in our study. In the Section 3.5. “antioxidant activity”, Tmp really showed good removal efficiency on hydroxyl and presented a dose-dependent effect. The antioxidant activity did not start reaching a “plateau” after 0.1 mg/ml, it continuously increased but it just increased slowly in this range from 0.1 mg/ml to 0.4 mg/ml. When the concentrations of Tmp were 0.1 mg/ml, 0.2 mg/ml and 0.4 mg/ml, the corresponding hydroxyl scavenging capacity of Tmp were 52.43%, 53.68% and 55.16%.      

(III) Conclusions

I would suggest that authors:

  • highlight in this section the most novel observation of their work (e.g what is the finding not earlier reported in this scientific field);
  • make a short reference to follow up work on this scientific field (e.g any need for further in vitro/in vivo studies for optimisation)

Response: Thanks for the referee's helpful suggestion. As suggested by the referee, we have adjusted the section of Conclusions as follows:

In this study, a novel and homogeneous polysaccharide Tmp from T. matsutake was obtained and the structural features were characterized. Tmp was mainly consisted of Ara, Man, Glc, and Gal with a molar ratio of 1.9:13.6:42.7:28.3, respectively. The main structure of Tmp was composed of →2,5)-α-L-Araf-(1→, →3,5)-α-L-Araf-(1→, β-D-Glcp-(1→, α-D-Manp-(1→, α-D-Galp-(1→, →4)-β-D-Galp-(1→, →3)-β-D-Glcp-(1→, →3)- α-D-Manp-(1→, →6)-3-O-Me-α-D-Manp-(1→, →6)-α-D-Galp-(1→, →3,6)-β-D-Glcp-(1→, →6)-α-D-Manp-(1→ residues. Moreover, Tmp showed good antioxidant activity, α-amylase and α-glucosidase inhibition activity. The inhibition kinetics analysis indicat-ed that Tmp exhibited an uncompetitive manner on α-amylase but played a mixed inhibi-tion effect on α-glucosidase. Hypoglycemic tests in IR-HepG2 cell mode revealed that Tmp possessed favorable hypoglycemic activity. Further study on hypoglycemic activity needs to be carried out in vivo studies for optimization.

References

  • Please do a quality check for formatting issues (e.g. italics for journal/bold for year?) Please check and correct according to the author’s guidelines (see the template below..)
  1. Author 1, A.B.; Author 2, C.D. Title of the article. Abbreviated Journal NameYearVolume, page range.

Response: Thanks for the referee's helpful suggestion. As suggested by the referee, we have checked the references and made changes according to author’s guidelines. 

Best regards,

Ying-Jie Zeng, Associate Professor, Corresponding author

College of Food Science & Technology,

Southwest Minzu University,

Wuhou District, Chengdu, China

E-mail: phhzyj@163.com

Cell: +86-15602408766
